# Estimating diagnostic uncertainty in artificial intelligence assisted pathology using conformal prediction

Henrik Olsson [1] ✉, Kimmo Kartasalo [1], Nita Mulliqi[1], Marco Capuccini[2], Pekka Ruusuvuori[3,4], Hemamali Samaratunga[5], Brett Delahunt[6], Cecilia Lindskog [7], Emiel A. M. Janssen[8,9], Anders Blilie[8,9], ISUP Prostate Imagebase Expert Panel, Lars Egevad[10], Ola Spjuth [2] & Martin Eklund [1]

Unreliable predictions can occur when an artificial intelligence (AI) system is presented with data it has not been exposed to during training. We demonstrate the use of conformal prediction to detect unreliable predictions, using histopathological diagnosis and grading of prostate biopsies as example. We digitized 7788 prostate biopsies from 1192 men in the STHLM3 diagnostic study, used for training, and 3059 biopsies from 676 men used for testing. With conformal prediction, 1 in 794 (0.1%) predictions is incorrect for cancer diagnosis (compared to 14 errors [2%] without conformal prediction) while 175 (22%) of the predictions are flagged as unreliable when the AI-system is presented with new data from the same lab and scanner that it was trained on. Conformal prediction could with small samples ($N = 49$ for external scanner, $N = 10$ for external lab and scanner, and $N = 12$ for external lab, scanner and pathology assessment) detect systematic differences in external data leading to worse predictive performance. The AI-system with conformal prediction commits 3 (2%) errors for cancer detection in cases of atypical prostate tissue compared to 44 (25%) without conformal prediction, while the system flags 143 (80%) unreliable predictions. We conclude that conformal prediction can increase patient safety of AI-systems.

There are good indications that artificial intelligence (AI) will transform healthcare and offer improved patient care at a reduced cost[1]. Radiology and pathology are likely to be the first fields in medicine where AI will be broadly implemented[2]. A barrier to the implementation of AI systems in healthcare is the need to ensure accurate AI performance across different settings[3–5]. Widespread application of AI systems will inevitably expose these systems to data beyond the domain upon which they were trained, either because it is unusual (e.g., atypical tissue previously unseen for the AI) or because it originates from a different imaging scanner provider, a different laboratory, or a different patient population. The ability to detect unreliable predictions will therefore be key to AI implementation in healthcare. Most AI

[1]Department of Medical Epidemiology and Biostatistics, Karolinska Institutet, Stockholm, Sweden. [2]Department of Pharmaceutical Biosciences, Uppsala University, Uppsala, Sweden. [3]Institute of Biomedicine, University of Turku, Turku, Finland. [4]Faculty of Medicine and Health Technology, Tampere University, Tampere, Finland. [5]Aquesta Uropathology and University of Queensland, Brisbane, QLD, Australia. [6]Department of Pathology and Molecular Medicine, Wellington School of Medicine and Health Sciences, University of Otago, Wellington, New Zealand. [7]Department of Immunology, Genetics and Pathology, Uppsala University, Uppsala, Sweden. [8]Department of Pathology, Stavanger University Hospital, Stavanger, Norway. [9]Faculty of Science and Technology, University of Stavanger, Stavanger, Norway. [10]Department of Oncology Pathology, Karolinska Institutet, Solna, Sweden. *A list of authors and their affiliations appears at the end of the paper. A list of members and their affiliations appears in the Supplementary Information. ✉e-mail: henrik.olsson@ki.se

systems and prediction models, however, only provide point predictions, without any associated assessment of how reliable a prediction is.

Conformal prediction (CP) is a mathematical framework that can be used together with any AI system or prediction model to guarantee the error rate is bounded by a pre-specified level[6]. Using CP, it is possible to set the desired confidence level, say 95%, in the prediction; the conformal predictor will then provide a prediction region around the point prediction that contains the true label with 95% probability (for classification problems, this prediction region corresponds to a set of labels, a *multilabel prediction*). If the prediction does not reach this confidence level, an *empty prediction* can be made, or, if the prediction region associated with a point prediction is too large for the prediction to be informative, the corresponding prediction can be flagged for human intervention. The conformal predictor can thus function as a quality control system for ensuring that only reliable predictions are made. See Supplementary Table 1 for a brief introduction to CP.

We and others have shown that AI-assisted pathology is a promising pathway to meet the challenges associated with histopathological diagnosis, for example, applied to the grading of prostate biopsies according to the International Society of Urological Pathology (ISUP) grading scheme[7–10]. Despite this, questions remain how about these models can generalize to data that differ from the training data.

In this study, we develop conformal predictors for AI-assisted prostate pathology. We show how these predictors can be used to detect unreliable predictions due to changes in tissue preparation techniques in different laboratories, digitization utilizing different digital pathology scanners, and the presence of atypical prostatic tissue, such as variants of prostatic adenocarcinoma and benign mimics of cancer. We believe this approach can have widespread utility in ensuring patient safety of clinically implemented AI systems.

## Results

We applied the CP framework to AI for the diagnosis and grading of prostate cancer in biopsies using data from the STHLM3 study (Fig. 1 and Supplementary Fig. 1). STHLM3 was a prostate cancer screening trial in men aged 50–69 years undertaken in Stockholm, Sweden, during 2012–2015 (ISRCTN84445406)[11]. For the training of the AI algorithm and conformal predictor, we included a digitized selection of 7788 formalin-fixed and hematoxylin and eosin-stained biopsies from 1192 STHLM3 participants. All slides were digitized using either Hamamatsu C9600-12 ($n = 5124$) or Aperio ScanScope AT2 scanners ($n = 2664$). Details of the selection and digitization of specimens have been described previously[7]. The underlying AI system was trained using convolutional deep neural networks following Ström et al.[7] To evaluate the CP framework, we assessed efficiency, defined as the fraction of all predictions resulting in a correct single-label prediction. We also assessed validity (the error rate), not exceeding the pre-specified significance level of the conformal predictor, for cancer detection and ISUP grading.

We employed a collection of six different datasets (numbered 1–6 below) comprising, in total, 3059 digitized biopsies for the evaluation

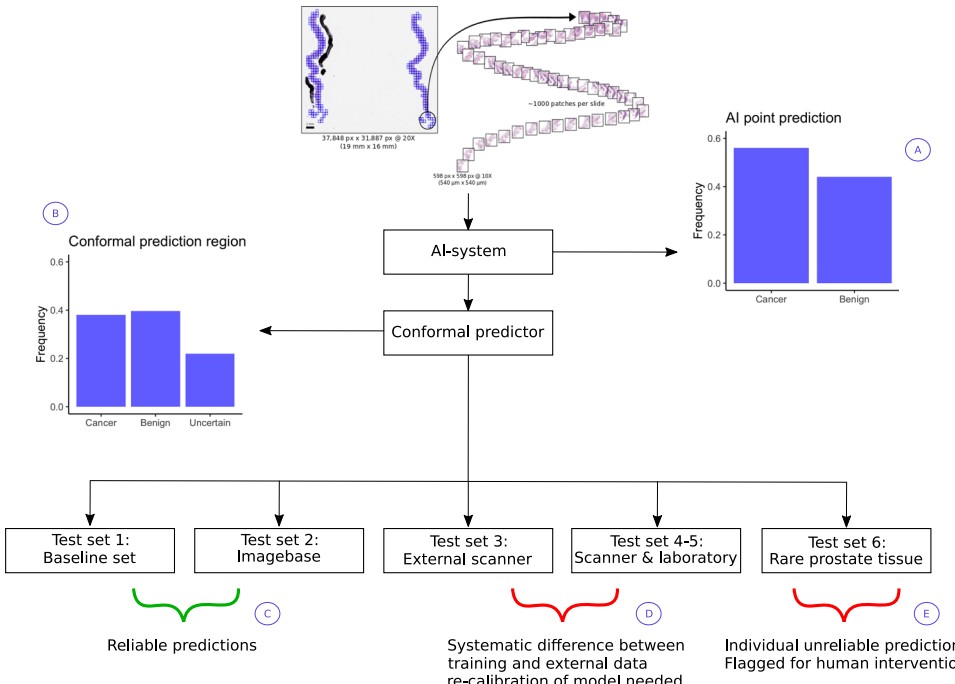

**Fig. 1 | Overview of study design. A** The AI system delivers point predictions, with no assessment of how reliable they are. **B** Conformal prediction is used to identify unreliable predictions from AI models. Unreliable predictions from AI systems can occur for several reasons, for example, because the AI system is presented with atypical data that it has not seen during training or because there are systematic differences between training data and data generated within the setting that the AI system is deployed. **C** Two datasets were utilized for the evaluation of AI cancer detection and ISUP grading performance under idealized conditions, where biopsies were processed in the same pathology laboratory and digitized using the same scanners as the biopsies in the training data. **D** The conformal predictor's ability to detect systematic differences between training data and external data was evaluated. Identification of such differences would then need to trigger corrective actions so that patient safety can be ensured. Two different data sources were used. In Test set 3, a different scanner was used for digitizing histopathology slides of prostate biopsies compared to the training data. In Test set 4 and Test set 5, both the pathology laboratory and the digital scanner that was used to prepare the dataset were different compared to the training data. **E** We evaluated the conformal predictor's ability to detect unusual prostate tissue that the AI model had limited or no previous exposure to during training. Test set 6 contained a set of 179 biopsies with atypical prostatic tissue, such as variants of prostatic adenocarcinoma and benign mimics of cancer. The conformal predictor should be able to flag atypical cases that are unsuitable for automated diagnosis, and that human intervention may be needed. ISUP: International Society of Urological Pathology. Imagebase database: a reference database developed by ISUP to promote the standardization of reporting urological pathology and containing cases independently reviewed by 23 highly experienced urological pathologists.

**Table 1 | Baseline characteristics of biopsy cores from the training data and the six different test datasets used for evaluation of the AI system and the conformal predictor**

| ISUP distribution of biopsies in training and test sets | | | | | | | |
|---|---|---|---|---|---|---|---|
| Cancer grade | Training sets | | Test sets | | | | | |
| | Deep neural network training set ($n = 6951$) | Conformal prediction calibration set ($n = 837$) | (1) Baseline test set ($n = 794$) | (2) Imagebase ($n = 87$) | (3) External scanner ($n = 449$) | (4) External scanner and external pathology laboratory ($n = 330$) | (5) External scanner and external pathology laboratory ($n = 1220$) | (6) Rare prostate tissue morphology ($n = 179$) |
| Benign | 3724 (54%) | 471 (56%) | 440 (55%) | 0 (0%) | 91 (20%) | 108 (33%) | 861 (71%) | 109 (61%) |
| ISUP 1 | 1530 (22%) | 176 (21%) | 172 (22%) | 21 (24%) | 183 (41%) | 65 (20%) | 206 (17%) | 51 (28%) |
| ISUP 2 | 539 (8%) | 80 (10%) | 62 (8%) | 32 (37%) | 64 (14%) | 63 (19%) | 61 (5%) | 19 (11%) |
| ISUP 3 | 263 (4%) | 35 (4%) | 31 (4%) | 15 (17%) | 33 (7%) | 49 (15%) | 45 (4%) | 0 (0%) |
| ISUP 4 | 469 (7%) | 51 (6%) | 41 (5%) | 8 (9%) | 47 (10%) | 19 (6%) | 22 (2%) | 0 (0%) |
| ISUP 5 | 426 (6%) | 24 (3%) | 48 (6%) | 11 (13%) | 31 (7%) | 26 (8%) | 25 (2%) | 0 (0%) |

The Imagebase dataset was independently graded by 23 uropathologists (the mode ISUP grade is shown in the table). ISUP: International Society of Urological Pathology. ISUP 1 (Gleason score 3 + 3), ISUP 2 (Gleason score 3 + 4), ISUP 3 (Gleason score 4 + 3), ISUP 4 (Gleason score 4 + 4, 3 + 5, and 5 + 3), ISUP 5 (Gleason score 4 + 5, 5 + 4, and 5 + 5).

of the AI system and conformal predictor. The baseline characteristics of biopsy cores from the training data and the six different datasets used for the evaluation of the AI system and the conformal predictor are shown in Table 1. All biopsies in both the training and Test set 1–4 and 6 were graded by Professor Lars Egevad (L.E.) according to the ISUP 2014 guidelines. The test set 2 was also independently graded by 22 additional uropathologists. The test set 5 was graded at the Department of Pathology, Stavanger University Hospital.

### Evaluation of AI performance under idealized conditions

Test set 1: As a baseline, we evaluated the AI system and the conformal predictor for undertaking cancer detection and ISUP grading on an independent dataset of 794 biopsies from 123 men from the STHLM3 study (no biopsies from men in the test set were included in the training data). This dataset was utilized for testing under idealized conditions, where biopsies were processed in the same pathology laboratory and digitized using the same scanners as the biopsies in the training data. The performance for the discrimination between benign biopsy cores and cores containing cancer on the baseline test set (Test set 1) was high; only one cancer case was erroneously classified as benign by the conformal predictor at a confidence level of 99.9% (Table 2). The overall efficiency—the percent correct single prediction out of all predictions—was 78% (72% for benign biopsy cores and 86% for biopsies with cancer). This means that the conformal predictor flagged 22% of predictions for human review. No empty set predictions were made. In contrast, the AI system without CP committed 14 (2%) errors for cancer detection (Table 2).

Agreement by two-thirds across a panel of experienced uropathologists has previously been defined as a consensus grade[12]. Consequently, we evaluated the efficiency of the conformal predictor at a 67% confidence level for ISUP grading on Test set 1: 98 (28%) errors were committed by the conformal predictor using the grade assigned by an experienced uropathologist (L.E.) as the gold standard (Table 2). There were 10 (3%) empty predictions and 72 (20%) multiple predictions, and the overall efficiency (correct single predictions) was 49%. For a confidence level of 80%, the error rate and the number of empty predictions were lower (62 errors [18%] and 7 empty predictions [2%]), while the number of multiple predictions was higher ($n = 153$ [43%]). The AI system without CP committed 117 (33%) errors for ISUP grading (Table 2). The conformal predictor produced lower class-wise error rates compared to the AI system without CP for ISUP grades 2–5, but the conformal predictor produced a higher error rate for ISUP 1 (28%) compared to the error rate by the AI system (15%). A large number of ISUP 1 training examples (47% of all cancer cases; Table 1) gives the AI system a relatively better performance for recognizing this class. This means that we can allow for higher confidence within these classes

when we apply the conformal predictor without causing unreasonably wide (and therefore non-informative) prediction intervals. Thus, we evaluated a mixed confidence approach, where a higher 85% confidence level was used for ISUP 1 and a confidence level of 67% was used for ISUP grades 2–5. Using this mixed-confidence approach, the AI system committed 70 (20%) errors (compared to 117 errors [33%] without CP), and the overall efficiency was 168 (47%) (Table 2).

Conformal predictors output multiple predictions in cases where they cannot assign reliable single predictions. It is, therefore, not possible to directly compare the sensitivity and specificity with and without the use of CP. However, we provide an experimental example, aiming to describe how a combination of CP coupled with human assessment could work to improve the accuracy of prostate pathology. In this example, we assume expert uropathologist-level diagnostic accuracy on the biopsies for which the conformal predictor has been identified as unreliable. The standalone AI system achieved an AUC of 99.7% for cancer detection, while the experimental approach, combining the point predictions from the AI, with an assessment of the CP regions, and human assessment of unreliable cases, achieved an AUC of 99.9% (Supplementary Fig. 5).

### Evaluation of AI performance against a panel of uropathologists

The test set 2: We compared the reliability of the AI system's prediction, as assessed by the conformal predictor, to that of expert uropathologists. We utilized 87 digitized biopsies from the Imagebase database, developed by ISUP, to promote the standardization of reporting of urological pathology and containing cases independently reviewed by 23 highly experienced urological pathologists (the Imagebase panel; Supplement Section S1). Using the mode of the ISUP grades assigned by the 23 Imagebase uropathologists as the gold standard, the overall efficiency was 33 and 49% for confidence levels 80 and 67%, respectively (Supplementary Table 2). The prediction regions covered 65% of the individual votes by the 23 pathologists at a 67% confidence level, and 83% of the panel votes at a confidence level of 80% (Supplementary Data 1). This means that the uncertainty estimated to be associated with the grades assigned by the AI closely approximates the uncertainty associated with the grading performed by different pathologists. The size of the prediction regions for the multi-label predictions was typically two ISUP grades (Supplementary Data 1 and Supplementary Table 3).

### Detection of systematic differences in external data

To test the conformal predictor's ability to identify systematic differences in test data when compared to training data, we used two datasets: Test set 3 involved the use of a scanner that differed from that used to prepare the training dataset. By exploiting the fact that

**Table 2 | Prediction regions on the baseline cases (Test set 1) for cancer detection and ISUP grading, n (%)**

| Confidence | | Benign (n = 440) | Cancer (n = 354) | All biopsies (n = 794) | | | |
|---|---|---|---|---|---|---|---|
| *Conformal prediction regions for cancer detection* | | | | | | | |
| 99.90% | Error, n (%) | 0 (0) | 1 (0) | 1 (0) | | | |
| | Empty, n (%) | 0 (0) | 0 (0) | 0 (0) | | | |
| | Single predictions, n (%) | 315 (72%) | 303 (86%) | 618 (78%) | | | |
| | Multiple predictions, n (%) | 125 (28%) | 50 (14%) | 175 (22%) | | | |
| *Conformal prediction regions for ISUP grading* | | | | | | | |
| | | ISUP 1 (n = 172) | ISUP 2 (n = 62) | ISUP 3 (n = 31) | ISUP 4 (n = 41) | ISUP 5 (n = 48) | All grades (n = 354) |
| 67% | Error, n (%) | 49 (28%) | 20 (32%) | 7 (23%) | 11 (27%) | 11 (23%) | 98 (28%) |
| | Empty, n (%) | 5 (3%) | 2 (3%) | 0 (0%) | 2 (5%) | 1 (2%) | 10 (3%) |
| | Single predictions, n (%) | 114 (66%) | 20 (32%) | 7 (23%) | 12 (29%) | 21 (44%) | 174 (49%) |
| | Multiple predictions, n (%) | 4 (2%) | 20 (32%) | 17 (55%) | 16 (39%) | 15 (31%) | 72 (20%) |
| 80% | Error, n (%) | 28 (16%) | 16 (26%) | 6 (19%) | 7 (17%) | 5 (10%) | 62 (18%) |
| | Empty, n (%) | 3 (2%) | 2 (3%) | 0 (0%) | 1 (2%) | 1 (2%) | 7 (2%) |
| | Single predictions, n (%) | 97 (56%) | 8 (13%) | 3 (10%) | 6 (15%) | 18 (38%) | 132 (37%) |
| | Multiple predictions, n (%) | 44 (26%) | 36 (58%) | 22 (71%) | 27 (66%) | 24 (50%) | 153 (43%) |
| *Conformal prediction regions for ISUP grading: Class-wise confidence levels* | | | | | | | |
| Class-wise confidence levels | | Confidence 85% for ISUP 1 | Confidence 67% for ISUP 2—ISUP 5 | | | | |
| | | ISUP 1 (n = 172) | ISUP 2 (n = 62) | ISUP 3 (n = 31) | ISUP 4 (n = 41) | ISUP 5 (n = 48) | All grades (n = 354) |
| | Error, n (%) | 20 (12%) | 20 (32%) | 7 (23%) | 12 (29%) | 11 (23%) | 70 (20%) |
| | Empty, n (%) | 2 (1%) | 2 (3%) | 0 (0) | 1 (2%) | 1 (2%) | 6 (2%) |
| | Single predictions, n (%) | 117 (68%) | 11 (18%) | 7 (23%) | 12 (29%) | 21 (44%) | 168 (47%) |
| | Multiple predictions, n (%) | 33 (19%) | 29 (47%) | 17 (55%) | 16 (39%) | 15 (31%) | 110 (31%) |
| *AI point predictions for cancer detection* | | | | | | | |
| | | Benign (n = 440) | Cancer (n = 354) | All biopsies (n = 794) | | | |
| | Error, n (%) | 4 (1%) | 10 (3%) | 14 (2%) | | | |
| | Correct, n (%) | 436 (99%) | 344 (97%) | 780 (98%) | | | |
| *AI point predictions for ISUP grading* | | | | | | | |
| | | ISUP 1 (n = 172) | ISUP 2 (n = 62) | ISUP 3 (n = 31) | ISUP 4 (n = 41) | ISUP 5 (n = 48) | All grades (n = 354) |
| | Error, n (%) | 26 (15%) | 33 (53%) | 19 (61%) | 21 (51%) | 18 (38%) | 117 (33%) |
| | Correct, n (%) | 146 (85%) | 29 (47%) | 12 (39%) | 20 (49%) | 30 (62%) | 237 (67%) |

The results are presented both as prediction regions by the conformal predictor and point predictions by the AI system without the conformal predictor. Cancer detection is reported at a confidence level of 99.9%, and ISUP grading is reported at 67% and 80% confidence levels, as well as using class-wise confidence levels (85% for ISUP 1 and 67% for ISUP 2–5). Labels are included in the prediction region if their confidence is higher than a user-specified desired confidence (e.g., 99.9%). The error is the fraction of true labels not included in the prediction region. A multi-label prediction means that the prediction is uncertain, and the model cannot distinguish between several possible class labels at the desired confidence. Empty set predictions are examples where the model could not assign any label, typically meaning that the example was very different from the data the model was trained on. *ISUP* International Society of Urological Pathology.

biopsies from the STHLM3 study were digitized using two different scanners, we trained the AI system only on Aperio images and evaluated this on a set of 449 slides scanned using both scanners, thus creating a paired dataset where a change in the scanner was the only variable to impact prediction performance. We then reversed the experiment and used only Hamamatsu images for training, with subsequent evaluation on the 449 images scanned both using Aperio and Hamamatsu scanners. Test set 4 included variations in both the laboratory and slide scanner. This set consisted of 330 slides processed and digitized at the Karolinska University Hospital, representing both a different laboratory and scanner compared to the training data. Test set 5 consisted of 1220 slides processed and digitized at the Stavanger University Hospital. These slides were used as an external

validation set to assess the conformal predictor's ability to detect systematic differences in test data.

Figure 2 shows the expected to observed cancer detection error rates at significance levels between 0 and 100% for Test sets 1, 3, 4, and 5. The conformal predictor was well calibrated for Test set 1 (Fig. 2, Panel A), when the same scanner was used for training and evaluation, but not for Test set 3 (external scanner; Fig. 2, Panel B and Supplementary Fig. 4), Test set 4 (external scanner and laboratory; Fig. 2, Panel C), or Test set 5 (the external set; Fig. 2, Panel D) (Kolmogorov–Smirnov $P < 0.05$ for all tests). The prediction regions were valid (Kolmogorov–Smirnov $P > 0.05$) when the same scanner was used for training and evaluation on Test set 3 (Supplementary Fig. 2). The number of observations needed to detect a systematic

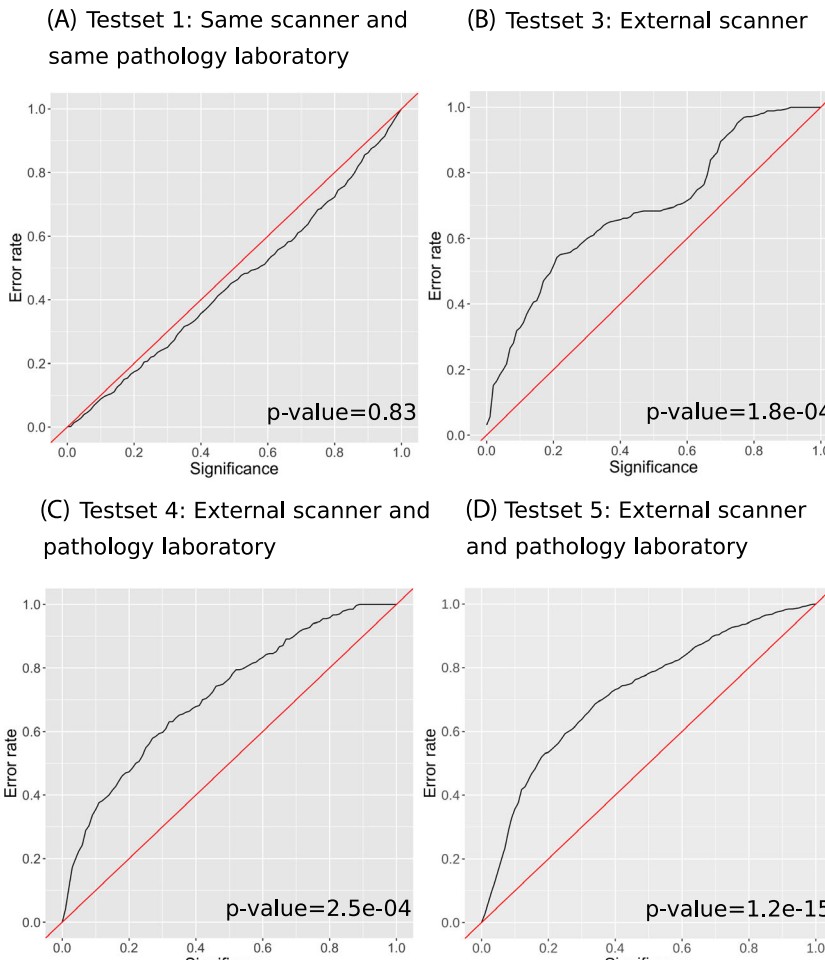

(A) Testset 1: Same scanner and same pathology laboratory

(B) Testset 3: External scanner

(C) Testset 4: External scanner and pathology laboratory

(D) Testset 5: External scanner and pathology laboratory

**Fig. 2 | Calibration plot of the observed prediction error (i.e., the fraction of true labels not included in the prediction region) on the $y$-axis and the pre-specified significance level $\varepsilon$ i.e., the tolerated error rate.** The conformal predictor is valid if the observed error rate does not exceed $\varepsilon$ i.e., the observed error rate should be close to the diagonal line, the tolerated error rate for all significance levels. The main advantage of conformal predictors is that they provide valid predictions when new examples are independent and identically distributed to the training examples. The graphs show results for Test set 1, 3, and 4, respectively. Panel **A** shows prediction regions on Test set 1, an independent test set consisting of 794 biopsies from 123 men from the STHLM3 study, all from the same laboratory as well as scanned on the same scanners as the training data. Panel **B** shows prediction regions on Test set 3 (external scanner), a set of 449 slides, held out from training, that was scanned on a different scanner than the training data. To evaluate Test set 3 (external scanner), we excluded images scanned on Hamamatsu from

training, leaving 2152 Aperio images for training. The prediction regions were non-valid when evaluated on the new scanner (Hamamatsu), as the prediction error is larger than the tolerated error for all significance levels. Panel **C** shows prediction regions on Test set 4, a set of 330 slides from an external clinical workflow, these slides were processed using a different laboratory and a different scanner compared to the training data. Panel **D** shows prediction regions on Test set 5, an external test set of 1220 slides from Stavanger University Hospital representing an external clinical workflow; these slides were processed using a different laboratory and a different scanner compared to the training data and used as prospective validation of the conformal predictor. We used the Kolmogorov–Smirnov test of equality of the distribution of the predictions in the calibration set and each test dataset to test the validity of the prediction regions. The null hypothesis was that the samples were drawn from the same distribution. A $p$-value of less than 5% was considered statistically significant (two-sided).

difference between the training data and Test set 3 was 49 observations, and the corresponding number for Test set 4 was 10 observations, and 12 observations for Test set 5 (Supplementary Fig. 2).

### Detection of atypical prostate tissue
The test set 6: Lastly, to assess the conformal predictor on unusual morphological patterns that the underlying AI model had limited or no previous exposure to during training, we used a set of 179 biopsies containing rare prostate tissue morphologies. These tissue sections contain morphologies that are typically difficult to diagnose or grade for pathologists, such as benign mimics of prostate cancer and rare prostate cancer subtypes (adenosis [$n = 36$], basal cell hyperplasia [$n = 21$], clear cell cribriform hyperplasia [$n = 3$], prostatic atrophy [$n = 37$], posatrophic hyperplasia [$n = 5$], Cowper's glands [$n = 6$], cancer of atrophic type [$n = 11$], foamy gland cancer [$n = 13$], prostatic

intraepithelial neoplasia (PIN-like) carcinoma [$n = 3$], pseudohyper-plastic cancer [$n = 41$], small-cell cancer [$n = 3$]). Figure 3 illustrates example pathology images for a sample of the rare prostate tissue subtypes from this dataset.

For Test set 6, the conformal predictor flagged 143 predictions (80%) for human intervention (i.e., uncertain multiple predictions) and committed 3 errors (2%). In contrast, the AI system without CP, providing diagnosis for all samples, committed 44 errors (25%) (Table 3).

## Discussion
The main barrier to the implementation of AI systems in healthcare is to ensure accurate AI performance across different settings. In this study, we have used AI for histopathological diagnosis and grading of prostate cancer to demonstrate how a system based on CP can be used

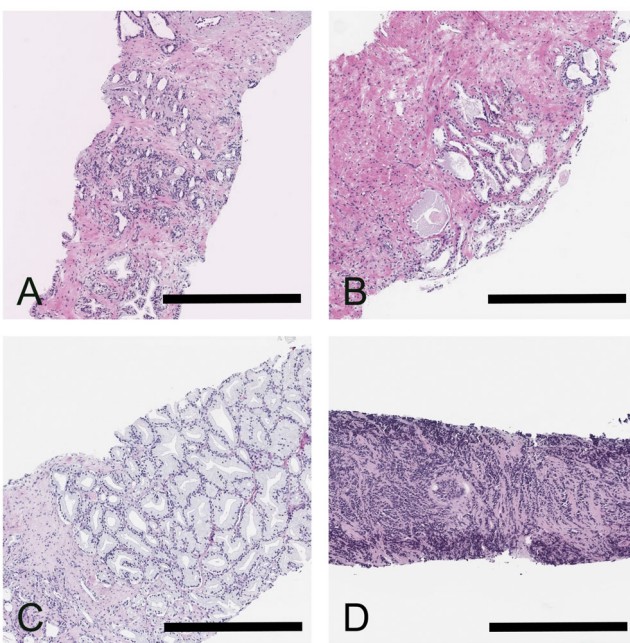

**Fig. 3 | Example pathology images of rare prostate tissue subtypes.** Panel **A** Benign prostatic tissue with postatrophic hyperplasia. Panel **B** Benign prostatic tissue with partial atrophy. Panel **C** Pseudohyperplastic cancer. Panel **D** Small cell carcinoma of the prostate. The case in Panel **B** was misinterpreted by AI as malignant, while the other cases were correctly classified as benign and malignant, respectively. Scale bars correspond to 500 μm.

to monitor predictions from AI systems and flag unreliable predictions for human intervention, as well as to identify systematic differences between the training data and external data leading to poor model performance.

Errors in prostate cancer diagnosis are rare but well recognized[13,14]. It is clear that there is less acceptance of machine learning mistakes than of human mistakes[15]. CP enables us to only accept predictions with high confidence, such that the error rate can be kept low. The tradeoff is that the conformal predictor can output empty or multiple predictions, which identifies cases where the conformal predictor cannot assign reliable predictions. In a scenario where a physician is assisted by an AI system, this provides important feedback to the physician so as to not rely on the AI system's predictions for these cases. In a scenario where the AI system would operate independently, such unreliable cases can be flagged for human inspection. Such a process enables synergisms between AI and humans, where the AI system may identify the clear-cut cases where its predictions are highly reliable, while expert pathologists have more time to focus on the challenging unreliable predictions (Table 2 and Supplementary Fig. 5). It is; however, important that the number of unreliable predictions does not become too high as this could lead to an unmanageable situation. In our results, we show that the unreliable (empty or multiple) predictions for prostate cancer detection were 22% at a confidence level of 99.9%. This represents an error rate of 0.1%, which is markedly lower than the 2% error rate that has been reported for pathologists[13,14]. Our results on Test set 6, containing unusual morphological patterns, demonstrate how CP can identify cases that the underlying AI system had little or no exposure to during training. Using CP, the errors could be decreased from 44 (25%) to 3 (2%) at the cost of flagging 143 predictions (80%) for human inspection.

The Gleason grading system suffers from a high level of inter-observer variability, making grading a challenging task for AI systems[16]. For example, using CP at a confidence level of 67% (representing the level of the agreement defined by the Imagebase panel to constitute consensus[12]), generated 49% single predictions and 20% multiple

**Table 3 | Cancer detection on rare prostate morphologies (Test set 6)**

| | Benign mimics of prostate cancer | | | | | | Rare prostate cancer subtypes | | | | | |
|---|---|---|---|---|---|---|---|---|---|---|---|---|
| Confidence 99.9% | Adenosis, n=36 | Basal cell hyperplasia, n=21 | Clearcell cribriform hyperplasia, n=3 | Prostatic atrophy, n=37 | Post-atrophic hyperplasia, n=5 | Cowper's glands, n=6 | Atrophy like cancer, n=11 | Foamy, n=13 | PIN like cancer, n=3 | Pseudo-hyperplastic, n=41 | Small-cell cancer, n=3 | All cases, n=179 |
| *Conformal prediction regions for cancer detection* | | | | | | | | | | | | |
| Error, n (%) | 0 (0) | 0 (0) | 0 (0) | 0 (0) | 0 (0) | 2 (33%) | 1 (9%) | 0 (0) | 0 (0) | 0 (0) | 0 (0) | 3 (2%) |
| Empty, n (%) | 0 (0) | 0 (0) | 0 (0) | 0 (0) | 0 (0) | 0 (0) | 0 (0) | 0 (0) | 0 (0) | 0 (0) | 0 (0) | 0 (0) |
| Single predictions, n (%) | 0 (0) | 0 (0) | 0 (0) | 0 (0) | 0 (0) | 0 (0) | 4 (36%) | 9 (69%) | 0 (0) | 19 (46%) | 1 (33%) | 33 (18%) |
| Multiple predictions, n (%) | 36 (100%) | 21 (100%) | 3 (100%) | 37 (100%) | 5 (100%) | 4 (67%) | 6 (55%) | 4 (31%) | 3 (100%) | 22 (54%) | 2 (67%) | 143 (80%) |
| *AI point predictions for cancer detection* | | | | | | | | | | | | |
| Error, n (%) | 16 (44%) | 1 (5%) | 0 (0) | 9 (24%) | 3 (60%) | 5 (83%) | 3 (27%) | 0 (0) | 2 (67%) | 4 (10%) | 1 (33%) | 44 (25%) |
| Correct, n (%) | 20 (56%) | 20 (95%) | 3 (100%) | 28 (76%) | 2 (40%) | 1 (17%) | 8 (73%) | 13 (100%) | 1 (33%) | 37 (90%) | 2 (67%) | 135 (75%) |

The results are presented both as prediction regions by the conformal predictor and point predictions by the AI system without the conformal predictor. The conformal prediction regions are reported at a confidence level of 99.9%. The error is the fraction of true labels not included in the prediction. A multi-label prediction means that the prediction is uncertain, and the model cannot distinguish between several possible class labels at the desired confidence. Empty set predictions are examples where the model could not assign any label, typically meaning that the example was very different from the data the model was trained on. These tissue sections contain morphologies that are typically difficult to grade for pathologists, including benign mimics of prostate cancer; adenosis, n=36, basal cell hyperplasia, n=21, clear cell cribriform hyperplasia, n=3, prostatic atrophy, n=37, postatrophic hyperplasia, n=5, Cowper's glands, n=6 and rare prostate cancer subtypes; atrophy like cancer, n=11, foamy gland, n=13, PIN-like cancer, n=3, pseudohyperplastic cancer, 41, small-cell cancer, n=3.

predictions. Rather than relying on a single assigned ISUP grade by a pathologist, we argue that the multiple predictions better reflect the true uncertainty in the grade assignment. This is further supported by the fact that prediction regions covered 65 and 83% (at 67 and 80% confidence levels, respectively) of the individual votes by the 23 pathologists on the Imagebase reference panel. These prediction regions had a median size of two ISUP grades, making them able to provide informative support for clinical decision-making.

CP relies on the assumption that the training data and the external data are received in random order from the same distribution. While this is often a reasonable assumption locally (e.g., patients from a given hospital's uptake area seeking care in random order), it is less likely to be true across different healthcare settings, where differences would be expected across patient populations, laboratories, and scanners. The validity property of CP enables it to identify if such systematic differences between training and external data lead to deteriorated AI performance, and warn that re-calibration of the underlying AI system is required to ensure the accuracy of predictions. In our results, 49 observations were required to identify that data from a new scanner resulted in worse performance, 10 observations were required when the data originated from an external lab and a new scanner, and 12 observations were required on the external set, using a 5% alpha level Kolmogorov–Smirnov test. More refined tests may lead to further improved test characteristics[17].

The problem of how quantifying the uncertainty of deep learning models has been addressed using different techniques, such as coupling deep learning models with Bayesian inference, Monte-Carlo dropout, or deep ensembles[18–20]. These methods are, however, hampered by additional computational costs and modeling complexity. In contrast, CP is lightweight to implement as part of any learning algorithm. Further, it is based on a well-defined mathematical framework that guarantees valid predictions (in fact, conformal predictors are essentially the only way to achieve valid prediction regions[6]), and the confidence levels can be varied to reflect the clinical cost of making an erroneous prediction.

The strengths of our study include the use of unique data sources that enabled us to study unusual morphologies, as well as systematic differences introduced by the use of different scanners and laboratories. All biopsies were graded by the same experienced pathologist (L.E.), thus avoiding total confounding between the pathologist and external data from a new laboratory. This reduced the likelihood that systematic differences could be introduced into the data due to variable interpretations by different pathologists. A further strength is the use of Imagebase data (Test set 2) for the evaluation of performance against a panel of expert uropathologists.

A limitation of the study is that we have not performed prospective validation. Additionally, we have utilized a simple conformal predictor (Methods Section: CP). It is likely that more advanced conformal predictors would achieve smaller prediction regions and higher proportions of reliable predictions. In this study, we have investigated prostate pathology as an example of how CP can be used for quality assurance of medical AI systems. Naturally, the same issues arise in all applications of AI, and conformal predicting will likely have utility for most other applications to ensure patient safety as we move toward the clinical implementation of AI systems.

## Methods
### Conformal prediction
CP is a mathematical framework for making predictions at exact levels of confidence, based on past experiences and previously seen data (see Supplementary Table 1 for a brief introduction to CP). The output from a conformal predictor is a *prediction region*, i.e., a set of class labels (for classification problems) or an interval (for regression). This contrasts with the single-value prediction from regular prediction models. For example, for a binary classification problem, the possible prediction sets are {0}, {1}, {0, 1}, or the empty set. Labels are included in the prediction region if their confidence is higher than a user-specified desired confidence level (e.g., 95%). A smaller prediction region is more *efficient* (more informative). Preferably, we would like the prediction region to only contain a single predicted label. A multi-label prediction means that the prediction is uncertain, and the model cannot distinguish between several possible class labels at the desired confidence level. For example, neither the assignment of a biopsy core as benign nor as containing cancer can be made at the desired confidence level, meaning that the conformal predictor will output a prediction set containing *both* classes. Although such a prediction is not incorrect per se, it is inconclusive, and human intervention would be needed. Empty set predictions are examples where the model could not assign any label, typically meaning that the example was very different from the data the model was trained on. Higher desired confidence in the prediction leads to larger prediction regions (analogously to how higher desired confidence in parameter estimates lead to larger confidence intervals).

The main advantage of conformal predictors is that they are mathematically guaranteed to provide *valid* predictions when new examples are independent and identically distributed to the training examples. This means that the probability that the prediction region determined by the conformal predictor does not include the true label is mathematically guaranteed to be less than or equal to a user-set significance level (a proportion of acceptable errors)[6]. The validity property of CP enables it to diagnose systematic differences between training data and external data or drifts in the data over time, and signal that re-calibration of the underlying machine learning model is needed to guarantee valid predictions.

Conformal predictors are built on top of the underlying prediction algorithm, and the framework can therefore be applied to all prediction algorithms[21]. The idea of CP is that for every new example, we try every possible class label, and evaluate how each candidate class label conforms with the training examples. The intuition behind CP is that data less conforming with training data should lead to less certain predictions. The concept of conformity is captured by a *nonconformity score*. For classification tasks, a commonly used nonconformity score is one minus the predicted probability. However, more advanced nonconformity scores are possible, e.g., by including information from the distribution of predictor variables to define the nonconformity measure. When making a prediction using the conformal predictor, the nonconformity score is used to calculate a *p*-value for each possible class label by computing the proportion of observations (pairs of predictor vectors and assigned class label) with more extreme nonconformity scores. Assigned labels are included in the prediction region if the *p*-value is larger than 1-c, where c is the desired confidence (e.g., 95%) in the prediction.

CP was originally defined as an *online transductive* framework[6,22]. In the online transductive mode, all available data is used to calculate the conformity score for each new example to make predictions on, which makes it necessary to retrain the underlying machine learning model for every calibration example, as well as for every test example. While the transductive online framework is attractive in the sense that it uses all available data for every new prediction, it is often computationally prohibitive. In addition, many applications (in particular in medicine) are not amenable to an online setting. Instead, a fixed model is used, and updates to the model are introduced with relatively long intervals in an *inductive offline* framework. CP has therefore been extended to the inductive setting[21], where one model is built from a training set and then applied to a test set. Inductive CP is computationally more efficient than the transductive conformal predictors. For inductive conformal predictors, the training dataset has to be divided into a proper training set (for training the underlying machine learning model) and a calibration set, where the calibration set is used for tuning the conformal predictor.

The most basic implementation of conformal predictors guarantees the error rate on a population level (across all examples in a dataset). This means that the error rate can conceivably be lower within one subpopulation of the dataset and higher within another. For example, the error rate could be lower for one class label and higher for the other. *Mondrian CP* was developed to achieve the pre-defined error rate within the substrata of the population[23]. The idea is simple: Instead of applying the CP framework across the entire population, it is instead tuned within each substratum, which mathematically guarantees the desired error rate within each stratum.

For further reading about CP, see Alvarsson et al., which describes CP in a comprehensive but non-technical way when used in drug discovery applications[24]. See also one related publication by Wieslander et al., where CP is combined with deep learning for region segmentation of whole slide lung tissue, to assess the confidence of the segmented regions[25].

### Application of CP to AI for prostate pathology

We applied the CP framework to AI for the diagnosis and grading of prostate cancer in biopsies (Fig. 2 and Supplementary Fig. 1). The underlying AI system was trained using two ensembles of convolutional deep neural networks (DNN) following Ström et al.[7] The first ensemble performed binary classification of image patches into benign or malignant, while the second ensemble classified patches into Gleason patterns 3–5. Each ensemble consisted of 30 Inception V3 models pretrained on ImageNet[26,27]. The predicted probabilities for the Gleason pattern at each location of the biopsy core were aggregated into slide-level features. The aggregated slide-level features were used as predictors for training a gradient-boosted trees classifier to predict the presence of cancer and ISUP grade[28]. See Ström et al for a more detailed description of the training of the AI system. DNNs were implemented in Python (version 3.6.9) using TensorFlow (version 2.6.2), and Python interface for XGBoost (version 1.2.1). Conformal predictors can be constructed in different ways. Here, we implemented a Mondrian inductive conformal predictor on a class basis to guarantee the desired error rate within each class. For the inductive conformal predictor, we split the training data into a 90% proper training set and a 10% calibration set. The training set was used to train the AI system, whereas the calibration set was used for the construction of the conformal predictor *p*-values. We used the predicted probability of an example belonging to a given class as a nonconformity measure.

For the training of the AI algorithm and conformal predictor, we included a digitized selection of 7788 formalin-fixed and hematoxylin and eosin-stained biopsies from 1192 participants from the STHLM3 prostate cancer screening trial (ISRCTN84445406)[11]. These data were split into a proper training set of 6951 biopsies from 1069 men, a calibration set consisting of 837 biopsies from 123 men, and a test set of 794 biopsies from 123 men (Table 1, Supplementary Fig. 1). Furthermore, we included 3059 biopsies from 676 men used for testing.

The present study was approved by the Stockholm regional ethics committee (permits 2012/572-31/1, 2012/438-31/3, and 2018/845-32), the Regional Committee for Medical and Health Research Ethics (REC) in Western Norway (permits REC/Vest 80924, REK 2017/71). Informed consent was provided by the participants in the Swedish dataset. For the other datasets, informed consent was waived due to the usage of de-identified prostate specimens in a retrospective setting.

### Statistical analysis

We applied the CP framework to AI for the diagnosis and grading of prostate cancer in biopsies (Fig. 1 and Supplementary Fig. 1). The underlying AI system was trained using convolutional deep neural networks (DNN) following Ström et al.[7] Briefly, the AI system outputs probabilities of biopsy cores containing only benign tissue and containing cancer with a specific ISUP grade. We trained two types of

models: *Model type 1*. In the first model, we used the entire training set of 7788 biopsies, where 6951 were used for training the DNN and 837 for calibrating the conformal predictor. This model was used on Test sets 1–2 and 4–5. *Model type 2*. To evaluate Test set 3 (external scanners), we first excluded images scanned on Hamamatsu from training, leaving 2152 images scanned on Aperio for training, and 449 images scanned on Hamamatsu for testing. We then reversed the experiment and used the 4078 images scanned on Hamamatsu for training, and 449 images scanned on Aperio for testing.

To evaluate the CP framework under idealized conditions, we assessed efficiency, defined as the fraction of all predictions resulting in a correct single-label prediction. We also assessed validity (the error rate), not exceeding the prespecified significance level of the conformal predictor, for cancer detection and ISUP grading on the baseline cases and Imagebase datasets (Test set 1 and 2). For cancer detection, we report classification efficiency at a confidence level of 99.9%.

The Imagebase panel defined a consensus grade as an agreement by two-thirds of the panel members. As a consequence of this, we evaluated the efficiency of the conformal predictor at a 67% confidence level for ISUP grading. We also evaluated ISUP grading at an 80% confidence level and at class-specific confidence levels (85% for ISUP 1 and 67% for ISUP 2–5). This means that there is a greater certainty that the true label of a predicted case will be within the prediction region determined by the CP, but also means that there will be larger and potentially clinically less informative regions (that is, a larger proportion of the biopsies flagged as requiring human intervention). To compare the output from the conformal predictor with ISUP grades assigned by expert uropathologists in the Imagebase panel, we calculated the proportion of individual pathologist votes that were covered by the prediction regions (Test set 2).

To investigate the use of CP to detect systematic differences between the training and test data, we employed the validity property of CP. For test sets 1, 3, 4, and 5 (corresponding to the baseline test set [Test set 1], external scanner [Test set 3], and external laboratory and scanner [Test sets 4 and 5]), we plotted the error rate of the conformal predictor at all significance levels between 0 and 100%. If the underlying AI system is well-calibrated to perform predictions on the test data, this plot forms a straight line along the diagonal. We tested the validity of the prediction regions using the Kolmogorov-Smirnov test of equality for the distribution of the predictions in the calibration set and each test dataset. Further, we determined how many observations would be needed for the detection of systematic differences between the training and external test data. This was undertaken by estimating the power of the Kolmogorov–Smirnov test for datasets 1, 3, and 4 by repeated random sampling of sets of the increasing sizes of predictions from the validation datasets. A *p*-value of less than 5% was considered statistically significant (two-sided). No participants were excluded from the analyses; there was no missing data for the statistical analyses. No formal sample size calculation was performed. Methods and results are presented in line with the transparent reporting of a multivariable prediction model for individual prognosis or diagnosis (TRIPOD) statement[29].

R version 4.0.0 was used for the implementation of the conformal predictors and all statistical analyses (R Foundation for Statistical Computing, Vienna, Austria. URL https://www.R-project.org/).

### Reporting summary

Further information on research design is available in the Nature Portfolio Reporting Summary linked to this article.

## Data availability

All data underlying this article cannot be shared publicly for the privacy of individuals that participated in the STHLM3 diagnostic study. They can be made available through contact with M.E. under research

collaboration and data-sharing agreements. Source data are provided in this paper. Anonymized demo versions of the datasets that are used for the main analyses in the manuscript are available at https://github.com/heolss/Conformal_analyses, and in the Zenodo database, at https://doi.org/10.5281/zenodo.7147740. All other data is available in the Source Data and Supplementary Information files. Anonymized training data are available as part of the PANDA challenge (https://www.kaggle.com/c/prostate-cancer-grade-assessment and Bulten et al. Nature Medicine, 2022[10]). Source data are provided in this paper.

## Code availability

The conformal prediction was implemented in R (version 4.0.0). The code for the analysis is available at https://github.com/heolss/Conformal_analyses, and in the Zenodo database at https://doi.org/10.5281/zenodo.7147740. The code used for training the underlying deep learning models cannot be shared due to patient privacy concerns and due to ongoing collaborations with the industry to implement the code for clinical diagnostic use. The code for data processing, model training, and prediction of the underlying deep learning models was implemented using the details described in our publication by Ström et al. Lancet Oncology, 2020[7]. See the methods section "Application of conformal prediction to AI for prostate pathology" for a brief overview of how the code was applied to the datasets presented in the manuscript, and see Ström et al for a more detailed description of the training of the AI system. The major components of our work are available in open-source repositories. Deep learning networks were implemented in Python (version 3.6.9), using the Keras API of Tensorflow (version 2.6.2) (https://www.tensorflow.org); Tensorflow Object Detection API (https://github.com/tensorflow/models/tree/master/research/object_detection), and Python interface for XGBoost (version 1.2.1). The code used for training the underlying deep learning models can be made available through contact with ME under research collaboration and data-sharing agreements. The PANDA challenge (Bulten et al. Nature Medicine, 2022[10]) website also contains tutorials for how to start working with the data (https://www.kaggle.com/code/wouterbulten/getting-started-with-the-panda-dataset) as well as the code used by the competing teams (https://www.kaggle.com/competitions/prostate-cancer-grade-assessment/code).

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

## Acknowledgements

M.E. received funding from the Swedish Research Council (Vetenskapsrådet; 2020-00692), the Swedish Cancer Society (Cancerfonden; 21 1715 Pj), the Magnus Bergvall Foundation, Region Stockholm, Svenska Druidorden, Åke Wibergs Stiftelse, and Swedish e-Science Research Center (SeRC), Karolinska Institutet, and the Swedish Prostate Cancer Foundation (Prostatacancerförbundet). K.K. received funding from KAUTE Foundation, David and Astrid Hägelen Foundation, Oskar

Huttunen Foundation, and Orion Research Foundation. Role of the funder: The funder had no role in the design of the study; data collection, analysis, and interpretation of the data; the writing of the manuscript; and the decision to submit the manuscript for publication. We want to thank Tony Ström, Carin Cavalli-Björkman, Astrid Björklund, and Britt-Marie Hune for assistance with scanning and database support.

## Author contributions

Study concept and design: H.O., M.E., M.C., O.S., and K.K. Acquisition of data: H.O., M.E., L.E., B.D., H.S., C.L., A.B., E.A.M.J., and N.M.; Analysis and interpretation of data: H.O. and M.E.; Drafting the paper: H.O. and M.E.; Critical revision of the paper for important intellectual content: H.O., K.K., N.M., M.C., P.R., H.S., B.D., C.L., E.A.M.J., A.B., L.E., O.S., and M.E.; Obtaining funding: M.E. and K.K. Men in Stockholm County that participated in the STHLM3 diagnostic study and contributed with the clinical information that made this study possible.

## Funding

## Competing interests

M.E. has four patents (WO2013EP74259 20131120, WO2013EP74270 20131120, WO2018EP52473 20180201, and WO2013SE50554 20130516) related to prostate cancer diagnostics pending, and has patent applications licensed to Thermo Fisher Scientific, and is named on a pending patent (1900061-1) related to cancer diagnostics quality control. All other authors declare no competing interests. The corresponding author had full access to all the data in the study and had final responsibility for the decision to submit for publication.

## Additional information

## ISUP Prostate Imagebase Expert Panel

**Hemamali Samaratunga[5], Brett Delahunt[6] & Lars Egevad[10]**

