## [Peer Review File · Nature Communications]

REVIEWER COMMENTS

Reviewer #1 (Remarks to the Author): expertise in AI and digital pathology

This paper addresses an interesting and timely question: how AI systems perform when presented with out-of-domain examples during testing. This is a practical challenge for the deployment of AI systems in healthcare.

The positive points of this manuscript are the large training and testing cohorts (several thousands of patients in total, although some of the individual test sets are very small with <50 pats). Also, the task (prostate cancer detection) is a common benchmark task in this field, although it could be considered a "solved" problem, given that several expert-level algorithms on this problem have been published, and partly approved for diagnostic use.

My main concern is that for me that I find the statistical endpoints of the system hard to judge. In a diagnostic task, I am interested in positive predictive value (PPV) and negative predictive value (NPV) as well as sensitivity and specificity and F1 score (all of them at pre-specified thresholds). Can the authors provide these metrics obtained by their new system and compare them with SOTA? These pieces of information seem to be missing and should be reported in results, discussed in the discussion, and even mentioned in the abstract.

The authors write "This means that the conformal predictor flagged 22% predictions" but what does this mean exactly? How do I judge this? Would this system be better if it had flagged 12% of predictions, or 32%, or 100%? or 0%? How can I compare this to the SOTA? This needs to be explained better, and not just in the discussion section.

A major point is that the source codes do not seem to be available. This is a must and the authors should provide a github link or even better, a DOI, to their codes.

Also, adherence to relevant reporting guidelines (TRIPOD, STARD, etc.) must be added.

Finally, the manuscript would benefit from more and more educational figures to illustrate the concept, and to report the results. Also, the batch effect and domain shifts could be visualized better. In addition, some actual pathology image data, including common and uncommon patterns could and should be shown.

In summary, this is a timely question and the method is an interesting concepts, but the authors failed to convince me that this method really improves on the SOTA in terms of hard end points.

Reviewer #2 (Remarks to the Author): expertise in artificial intelligence

In this manuscript, the authors highlight the need to detect unreliable AI predictions for practical deployment in healthcare. Conformal prediction (CP) can ensure reliability for some predictions while flagging others for humans to review. This study demonstrates the application of conformal prediction for AI diagnosis of prostate pathology to successfully detect unreliable predictions. The conformal prediction introduction is well-written and informative.

The original deep learning model and dataset comes from Strom et al. 2020. The test sets were selected to analyze different scenarios (idealized conditions, systematic differences, and morphological differences). Overall, this is a valuable contribution to the literature demonstrating the usefulness of conformal prediction for medical AI applications. This represents a novel application of conformal prediction to the field of histopathology. One related publication (doi 10.1109/JBHI.2020.2996300) on the use of CP in region segmentation on whole slide scans, which shared an author (OS) with the current ms, should probably have been cited--although the application described was substantially different.

However, there a few concerns:

The external data experiments should make it more clear if the model used was only trained on

the single scanner (such as in the Figure 2 caption). It is noted in the methods that the single scanner model is used to evaluate only test set 3. It is consequently not clear if Figure 2 is showing evaluation of both models or just the single scanner model on all three test sets. Also, it would be helpful to see a similar analysis for a model trained on a dataset with both scanners, and also the reverse (training on Hamamatsu, evaluation on Aperio).

More details on the AI system used from Strom et al. 2020 should be provided in the methods. Right now, the manuscript simply cites the paper without any additional details (Line 309). A simple summary mentioning the architectures used, training methodology, etc. should suffice. Also, I think a reference to a previous paper authored in part by one of the authors on a related application of conformal prediction could have been included. While it discusses the use of CP in analysis of whole slide images, the actual use case is quite different. (<https://doi.org/10.1109/JBHI.2020.2996300>)

The language is understandable. One edit is suggested: (Line 46): A barrier to the implementation of AI systems in healthcare is [add: "the need"] to ensure accurate AI performance across different settings.

Minor error: Table 1 has a mix of proportions and percentages

Reviewer #3 (Remarks to the Author): expert in prostate cancer pathology

Artificial intelligence (AI) has been proposed to be a very powerful tool in healthcare, especially in pathology and disease diagnosis. However, whether a specific AI system maintains a high accurate rate across different settings, or in another word, how to judge the reliability of AI prediction remain essential questions before a wide usage of the AI system in pathology. In this study, authors tried to utilize conformal prediction to assess the prediction quality of AI in the histopathological diagnosis and grading of prostate cancer. They firstly used slides of prostate cancer biopsies from the STHLM3 cohort for training and a small subset of slides in the same study for testing. They showed that conformal prediction was able to identify 0.1% error and 22% AI prediction as unreliable when the AI-system was exposed to new histological images from the same histopathological lab and slide scanner. Furthermore, they found that conformal prediction could detect systematic differences when AI was presented with external histopathological data. In addition, the AI-system with conformal prediction can flag atypical prostate tissue with a higher efficiency than AI without conformal prediction. This study is dealing with an important scientific question and provide a plausible solution to assess the accuracy of AI-assisted histopathological diagnosis. However, the authors should design/conduct additional study to clarify the following issues before publication in Nature Communications.

Here are my concerns and suggestions:

(1) The STHLM2 cohort is a very large cohort with 7406 patients and 59159 slides. Only 1192 patients and 7788 slides were used for training. Please provide the procedure and rational to select those slides for training.

(2) The training data are from the same cohort. As a matter of fact, the corresponding author has published several papers using multi-center PCa cohorts. It will be more convincing if this study can expand to other large cohorts (including different human races) for both training and testing.

(3) The n number for the atypical prostate tissues is small. The authors are suggested to include a larger number of atypical prostate tissue slides or to include PCa variants such as neuroendocrine prostate cancer or prostate cancer with neuroendocrine differentiation to test the application of conformal prediction.

(4) Conformal prediction is not the only solution to assess the uncertainty of AI. The authors are suggested to compare or at least discuss different methods to demonstrate the advantage of their conformal prediction in AI-assisted histopathological diagnosis.

(5) Please specify LE for the first time mentioned in the manuscript in line 93

RESPONSE TO REVIEWER COMMENTS

Reviewer #1 (Remarks to the Author): expertise in AI and digital pathology

This paper addresses an interesting and timely question: how AI systems perform when presented with out-of-domain examples during testing. This is a practical challenge for the deployment of AI systems in healthcare.

We thank the reviewer for this comment; indeed, we agree that this is a timely topic. As implementation of AI systems in healthcare get more common, we need to address the question of how to ensure that these systems are safe and provide value to patients in constantly changing clinical settings and across different populations, technical platforms, labs, etc.

The positive points of this manuscript are the large training and testing cohorts (several thousands of patients in total, although some of the individual test sets are very small with <50 pats). Also, the task (prostate cancer detection) is a common benchmark task in this field, although it could be considered a "solved" problem, given that several expert-level algorithms on this problem have been published, and partly approved for diagnostic use.

We agree with the reviewer that AI systems show promise for the diagnosis and grading of prostate biopsies. We and others have previously reported on the accuracy of state-of-the-art AI systems for prostate pathology (see e.g. our publications Ström et al. *Lancet Oncology*, 2020 [1] and Bulten et al. *Nature Medicine*, 2022 [2]), and demonstrated that the performance of these AI systems is comparable to that of international experts in prostate pathology with respect to sensitivity, specificity and grading concordance with expert uro-pathologists.

Despite these successes in advancing AI for prostate pathology, we caution against considering diagnostics and – in particular – grading to be solved problems. As far as we are aware, there is currently no high-level evidence demonstrating that AI systems improve quality of prostate pathology in a prospective clinical setting and we do not know of any ongoing prospective multi-site clinical trials. Similarly, to the best of our knowledge, only one study exists to this date, that independently assesses the validity of multiple algorithms for the problem in a multinational setting (the PANDA challenge [2], which we organized together with colleagues from Radboud MC and Google).

In particular, the problem with generalizability of AI systems is currently unsolved. Widespread clinical implementation of AI systems will inevitably expose these systems to data beyond the domain upon which they were trained, either because it is unusual or because it originates from a different imaging scanner provider, a different laboratory (or even stain variation or changing processes *within* a lab [3]), a different patient population, wear and tear of scanners, etc. These are challenging problems already with simple clinical

risk calculators or nomograms used today [4], and the challenges will only become larger with implementation of complex AI systems [5].

1. P. Ström, K. Kartasalo, H. Olsson, L. Solorzano, B. Delahunt, D. Berney, D. Bostwick, A. Evans, P. Humphrey, K. Iczkowski, J. Kench, G. Kristiansen, T. van der Kwast, K. Leite, J. McKenney, J. Oxley, C. Pan, H. Samaratunga, J. Srigley, H. Takahashi, T. Tsuzuki, M. Varma, M. Zhou, J. Lindberg, C. Lindskog, P. Ruusuvoori, C. Wählby, H. Grönberg, M. Rantalainen, L. Egevad, and **M. Eklund**. Artificial intelligence for diagnosis and grading of prostate cancer in biopsies: a population-based, diagnostic study. *Lancet Oncology*. 2020, 21(2):222-232. PMID: 31926806
2. W. Bulten, K. Kartasalo, P.C. Chen, P. Ström, H. Pinckaers, K. Nagpal, Y. Cai, D.F. Steiner, H. van Boven, R. Vink, C. Hulsbergen-van de Kaa, J. van der Laak, M.B. Amin, A.J. Evans, T. van der Kwast, R. Allan, P.A. Humphrey, H. Grönberg, H. Samaratunga, B. Delahunt, T. Tsuzuki, T. Häkkinen, L. Egevad, M. Demkin, S. Dane, F. Tan, M. Valkonen, G.S. Corrado, L. Peng, C.H. Mermel, P. Ruusuvoori, G. Litjens, and **M. Eklund** for the PANDA challenge consortium. International Assessment of Artificial Intelligence for Diagnosis and Gleason Grading of Prostate Cancer in Biopsies: The PANDA Challenge. *Nature Medicine*, 2022, 28(1):154-163. PMID: 35027755
3. C. Cong et al. Colour adaptive generative networks for stain normalisation of histopathology images. *Medical Image Analysis*, Volume 82, November 2022, 102580.
4. J. Chandra Engel, T. Palsdottir, D. Ankerst, S. Remmers, A. Mortezaei, V. Chellappa, L. Egevad, H. Grönberg, **M. Eklund**, and T. Nordström. External validation of the Prostate Biopsy Collaborative Group Risk Calculator (PBCG-RC) and the Rotterdam Prostate Cancer Risk Calculator (RPCRC) in a Swedish population-based screening cohort. *Eur Urol Open Sci*. 2022, 41:1-7. PMID: 35813248
5. S.G. Finlayson et al. Adversarial attacks on medical machine learning. *Science*. 2019 Mar 22;363(6433):1287-1289. PMID: 30898923

My main concern is that for me that I find the statistical endpoints of the system hard to judge. In a diagnostic task, I am interested in positive predictive value (PPV) and negative predictive value (NPV) as well as sensitivity and specificity and F1 score (all of them at pre-specified thresholds). Can the authors provide these metrics obtained by their new system and compare them with SOTA? These pieces of information seem to be missing and should be reported in results, discussed in the discussion, and even mentioned in the abstract.

We agree that the performance metrics the reviewer mentions are the conventional way to estimate the accuracy of prediction models on external test data. We and others have previously reported on the operating characteristics of state-of-the-art AI systems for prostate pathology (see e.g. our publications Ström et al. *Lancet Oncology*, 2020 [1] and Bulten et al. *Nature Medicine*, 2022 [2]), and demonstrated that the performance of these AI systems is comparable to that of international experts in prostate pathology with respect to sensitivity, specificity and grading concordance with expert uro-pathologists. The main goal of this study was however not to directly improve the accuracy of the performance of the underlying deep learning algorithm per se, but to construct a framework based on conformal prediction that can assess the reliability and estimate the uncertainty of the predictions for AI systems in digital pathology, such that unreliable predictions can be identified for human intervention.

Conformal predictors are primarily evaluated based on *validity* and *efficiency*. The validity (the maximum allowed error rate) refers to the calibration of the predictions. This is usually evaluated using calibration curves where the accuracy of the predictor is plotted against the desired confidence (Figure 3 in our manuscript). We evaluated the validity graphically using calibration curves and tested the validity using the Kolmogorov-Smirnov test. The efficiency is most commonly assessed by the width of the prediction intervals for regression or by the fraction of single label predictions for classification. The efficiency quantifies how informative or specific the prediction is, while still being valid. To evaluate the conformal prediction framework, we assessed efficiency, defined as the fraction of all predictions resulting in a correct single label prediction (Table 2 and 3).

However, as the reviewer probably is alluding to the, using conformal prediction can of course also affect the performance of the entire prediction system (consisting of the underlying deep learning algorithm plus the conformal predictor) by enabling flagging of unreliable predictions (as either multiple predictions or empty predictions). The errors that the AI system without conformal prediction is committing can thus be compared to the errors the AI together with conformal prediction is committing. In the previous version of the manuscript, we had included this information for the evaluation of Test set 5 (Table 3). We agree with the reviewer that this data should also have been included for Test set 1 (Table 2) for easier comparison of the impact of conformal predictor on the accuracy of the predictions. Table 2 has now been updated accordingly, and we have included a comparison of the results in the body of the manuscript (pages 6 and 7, line 108-148). Following the reviewer's suggestion, we have also updated the abstract to reflect this comparison. We note that Nature Communication abstracts are only 150 words long, making it challenging to include too many results in the abstract; we are however happy to make the abstract more descriptive and longer in case the editors feel it is needed.

Conformal predictors output multiple predictions in the cases where it cannot assign reliable single predictions. So, for example, in the case of classifying cases as either benign or malignant, the conformal predictor would classify an unreliable prediction as both classes (i.e. both malignant and benign). Although such a prediction is not incorrect per se, it is inconclusive and human intervention would be needed. It is therefore not possible to *directly* compare the sensitivity and specificity with and without the use of conformal prediction. However, this opens up for a very interesting discussion about the synergies of humans and machines working together to improve accuracy of prostate pathology, where the conformal predictor flags unreliable predictions for human assessment. We can then achieve expert uro-pathologist level diagnostic accuracy on the flagged biopsies, enabling us to construct ROC curves corresponding to the performance of AI system working together with expert human pathologists. We have now updated the manuscript to include such ROC curves in the supplement (Figure S5 and manuscript body page 7, line 139-148, and page 9, line 222-224).

The reporting of the positive and negative predictive values (PPV and NPV) and F1 score are complicated by the study design. The dataset used in the current manuscript was digitized for our paper published in the Lancet Oncology in 2020 [1]. For that work, we oversampled high-grade disease in order to cost- and time-efficiently ensure to get enough training data

of higher grades. A consequence of this design is that the interpretation of an estimated PPV is not straightforwardly meaningful in the context of a specific study population, since the PPV is a function of the prevalence in the dataset.

1. P. Ström, K. Kartasalo, H. Olsson, L. Solorzano, B. Delahunt, D. Berney, D. Bostwick, A. Evans, P. Humphrey, K. Iczkowski, J. Kench, G. Kristiansen, T. van der Kwast, K. Leite, J. McKenney, J. Oxley, C. Pan, H. Samaratunga, J. Srigley, H. Takahashi, T. Tsuzuki, M. Varma, M. Zhou, J. Lindberg, C. Lindskog, P. Ruusuvoori, C. Wählby, H. Grönberg, M. Rantalainen, L. Egevad, and **M. Eklund**. Artificial intelligence for diagnosis and grading of prostate cancer in biopsies: a population-based, diagnostic study. *Lancet Oncology*. 2020, 21(2):222-232. PMID: 31926806
2. W. Bulten, K. Kartasalo, P.C. Chen, P. Ström, H. Pinckaers, K. Nagpal, Y. Cai, D.F. Steiner, H. van Boven, R. Vink, C. Hulsbergen-van de Kaa, J. van der Laak, M.B. Amin, A.J. Evans, T. van der Kwast, R. Allan, P.A. Humphrey, H. Grönberg, H. Samaratunga, B. Delahunt, T. Tsuzuki, T. Häkkinen, L. Egevad, M. Demkin, S. Dane, F. Tan, M. Valkonen, G.S. Corrado, L. Peng, C.H. Mermel, P. Ruusuvoori, G. Litjens, and **M. Eklund** for the PANDA challenge consortium. International Assessment of Artificial Intelligence for Diagnosis and Gleason Grading of Prostate Cancer in Biopsies: The PANDA Challenge. *Nature Medicine*, 2022, 28(1):154-163. PMID: 35027755

The authors write "This means that the conformal predictor flagged 22% predictions" but what does this mean exactly? How do I judge this? Would this system be better if it had flagged 12% of predictions, or 32%, or 100%? or 0%? How can I compare this to the SOTA? This needs to be explained better, and not just in the discussion section.

We thank the reviewer for this comment; we acknowledge that the fraction of 22% predictions flagged for human intervention at a confidence level of 99.9% is somewhat hard to assess on its own. As described above, we have extended Table 2 to include results without the use of conformal prediction to facilitate easier interpretation of the results (see also pages 6 and 7, line 108-148 in the manuscript).

Conformal predictors are mathematically guaranteed to provide valid predictions when new examples are independent and identically distributed to the training examples. This means that the probability that the prediction region determined by the conformal predictor does not include the true label is mathematically guaranteed to be less than or equal to a user set significance level (a proportion of acceptable errors). At a given (user-defined) confidence level (in our case 99.9% for cancer classification, representing about one twentieth of errors committed by pathologists [1, 2]), we would like the number of unreliable predictions (i.e. those flagged for human intervention) to be as few as possible.

We believe that the use of conformal prediction helps to facilitate responsible implementation of an AI systems in the clinics, promoting patient safety by keeping the error rate low and providing ways to detect unreliable predictions.

1. Beltran, L. et al. Histopathologic False-positive Diagnoses of Prostate Cancer in the Age of Immunohistochemistry. *Am. J. Surg. Pathol.* 43, 361–368 (2019).

2. Oxley, J. D. & Sen, C. Error rates in reporting prostatic core biopsies. *Histopathology* 58, 759–765 (2011).

A major point is that the source codes do not seem to be available. This is a must and the authors should provide a github link or even better, a DOI, to their codes.

We fully agree with the reviewer of the importance of open science. The code for the implementation of the conformal predictor – the main contribution of the current study and the code that was developed specifically for this project – resides in the github repository https://github.com/heolss/Conformal_analyses. We include a Nature “Code and Software Submission Checklist” with this revision.

The code used for training the underlying deep learning models has a large number of dependencies on internal tooling, infrastructure and hardware, and its release is therefore not feasible. However, all experiments and implementation details are described in sufficient detail in our publication Ström et al. *Lancet Oncology*, 2020 [1] to support replication with non-proprietary libraries. Several major components of our work are available in open source repositories: Tensorflow (<https://www.tensorflow.org>); Tensorflow Object Detection API (https://github.com/tensorflow/models/tree/master/research/object_detection).

In addition, as part of the PANDA challenge (<https://www.kaggle.com/c/prostate-cancer-grade-assessment> and Bulten et al. *Nature Medicine*, 2022), we publicly released all training data used for the current study (as the test data was collected outside a trial setting without informed consent for data sharing, we are not permitted to release that data). The PANDA challenge website also contains tutorials for how to start working with the data (<https://www.kaggle.com/code/wouterbulten/getting-started-with-the-panda-dataset>) as well as the code used by the competing teams (<https://www.kaggle.com/competitions/prostate-cancer-grade-assessment/code>).

1. P. Ström, K. Kartasalo, H. Olsson, L. Solorzano, B. Delahunt, D. Berney, D. Bostwick, A. Evans, P. Humphrey, K. Iczkowski, J. Kench, G. Kristiansen, T. van der Kwast, K. Leite, J. McKenney, J. Oxley, C. Pan, H. Samaratunga, J. Srigley, H. Takahashi, T. Tsuzuki, M. Varma, M. Zhou, J. Lindberg, C. Lindskog, P. Ruusuvaori, C. Wählby, H. Grönberg, M. Rantalainen, L. Egevad, and **M. Eklund**. Artificial intelligence for diagnosis and grading of prostate cancer in biopsies: a population-based, diagnostic study. *Lancet Oncology*. 2020, 21(2):222-232. PMID: 31926806

Also, adherence to relevant reporting guidelines (TRIPOD, STARD, etc.) must be added.

The statistical analysis section has been updated with a statement that methods and results are reported according to the TRIPOD statements. A TRIPOD checklist is also included with this revision, which describes where the different sections of the TRIPOD statements can be found in the manuscript.

Finally, the manuscript would benefit from more and more educational figures to illustrate the concept, and to report the results. Also, the batch effect and domain shifts could be

visualized better. In addition, some actual pathology image data, including common and uncommon patterns could and should be shown.

We have added a new figure (Figure 1) to conceptually describe and illustrate conformal prediction. We have also added pathology imaging data (Figure 4). The conformal prediction calibration plots (Figures 3, S2, and S4) are a natural and standard way when using conformal prediction of how to illustrate batch effects or domain shifts (see e.g. Vovk et al. [1] and other publications on conformal prediction). To comply with the previous literature on conformal prediction, we argue that we should stick to this method of illustration.

1. Vovk, V., Gammerman, A. & Shafer, G. Algorithmic learning in a random world. *Algorithmic Learn. a Random World* 1–324 (2005). doi:10.1007/b106715

In summary, this is a timely question and the method is an interesting concepts, but the authors failed to convince me that this method really improves on the SOTA in terms of hard end points.

We thank the reviewer for the thorough review and the many good suggestions for how to improve our manuscript. We hope and believe that our answers to the comments above as well as the changes made to the manuscript addresses the reviewer's concerns.

Reviewer #2 (Remarks to the Author): expertise in artificial intelligence

In this manuscript, the authors highlight the need to detect unreliable AI predictions for practical deployment in healthcare. Conformal prediction (CP) can ensure reliability for some predictions while flagging others for humans to review. This study demonstrates the application of conformal prediction for AI diagnosis of prostate pathology to successfully detect unreliable predictions. The conformal prediction introduction is well-written and informative.

The original deep learning model and dataset comes from Strom et al. 2020. The test sets were selected to analyze different scenarios (idealized conditions, systematic differences, and morphological differences). Overall, this is a valuable contribution to the literature demonstrating the usefulness of conformal prediction for medical AI applications. This represents a novel application of conformal prediction to the field of histopathology. One related publication (doi 10.1109/JBHI.2020.2996300) on the use of CP in region segmentation on whole slide scans, which shared an author (OS) with the current ms, should probably have been cited--although the application described was substantially different.

We thank the reviewer for the nice words about this work.

However, there a few concerns:

The external data experiments should make it more clear if the model used was only trained on the single scanner (such as in the Figure 2 caption). It is noted in the methods that the single scanner model is used to evaluate only test set 3. It is consequently not clear if Figure 2 is showing evaluation of both models or just the single scanner model on all three test sets. Also, it would be helpful to see a similar analysis for a model trained on a dataset with

both scanners, and also the reverse (training on Hamamatsu, evaluation on Aperio).

We agree that the label of Figure 2 was not sufficiently clear. We have updated Figure 2 label to describe better that a single scanner model was used to evaluate on Test set 3.

Additionally, as suggested by the reviewer, we have also reversed the analysis with training on Hamamatsu and evaluation on Aperio. Supplementary Figure S4 shows results from that experiment. The results were nearly identical to the evaluation of Test set 3 currently in the manuscript, showing that the conformal predictor was able to identify systematic differences in test data compared to training data (in this case a new scanner). Please also see added sentences about this in the manuscript (page 8, line 171-173, and page 15, line 397-399).

More details on the AI system used from Strom et al. 2020 should be provided in the methods. Right now, the manuscript simply cites the paper without any additional details (Line 309). A simple summary mentioning the architectures used, training methodology, etc. should suffice.

We agree that more details about the underlying deep learning model would be beneficial. The methods section has now been updated with more information describing the training of the AI system (page 14, line 362-379).

Also, I think a reference to a previous paper authored in part by one of the authors on a related application of conformal prediction could have been included. While it discusses the use of CP in analysis of whole slide images, the actual use case is quite different. (<https://doi.org/10.1109/JBHI.2020.2996300>)

We thank the reviewer for this suggestion; the reference has been added to the manuscript.

The language is understandable. One edit is suggested: (Line 46): A barrier to the implementation of AI systems in healthcare is [add: "the need"] to ensure accurate AI performance across different settings.

The section has been updated with this suggestion.

Minor error: Table 1 has a mix of proportions and percentages

Thank you for pointing this out. The table has been updated accordingly.

Reviewer #3 (Remarks to the Author): expert in prostate cancer pathology

Artificial intelligence (AI) has been proposed to be a very powerful tool in healthcare, especially in pathology and disease diagnosis. However, whether a specific AI system maintains a high accurate rate across different settings, or in another word, how to judge the reliability of AI prediction remain essential questions before a wide usage of the AI system in pathology.

We could not agree more with this statement.

In this study, authors tried to utilize conformal prediction to assess the prediction quality of AI in the histopathological diagnosis and grading of prostate cancer. They firstly used slides of prostate cancer biopsies from the STHLM3 cohort for training and a small subset of slides in the same study for testing. They showed that conformal prediction was able to identify 0.1% error and 22% AI prediction as unreliable when the AI-system was exposed to new histological images from the same histopathological lab and slide scanner. Furthermore, they found that conformal prediction could detect systematic differences when AI was presented with external histopathological data. In addition, the AI-system with conformal prediction can flag atypical prostate tissue with a higher efficiency than AI without conformal prediction. This study is dealing with an important scientific question and provide a plausible solution to assess the accuracy of AI-assisted histopathological diagnosis. However, the authors should design/conduct additional study to clarify the following issues before publication in Nature Communications.

We thank the reviewer for the nice words about our study. Below we give answers to the comments raised by the reviewer, and detail the changes we have done to the manuscript.

Here are my concerns and suggestions:

(1) The STHLM2 cohort is a very large cohort with 7406 patients and 59159 slides. Only 1192 patients and 7788 slides were used for training. Please provide the procedure and rationale to select those slides for training.

The analysis of the current study was based on the digitized dataset that was used in Ström et al 2020, and the data collection for that study was performed in multiple rounds during 2017 to 2019. The data collection is described in more detail in Ström et al. Lancet Oncology 2020, and we have added a brief description to the current manuscript (page 5, line 84-88).

In summary, a selection of 8571 biopsies from 1289 STHLM3 study participants were randomly sampled (stratified random sampling within Gleason score to enrich for higher Gleason scores). The cases were chosen to represent the full range of diagnoses using random sampling. the majority of cases in a screening-by-invitation cohort such as the STHLM3 cohort are either benign or low-grade diseases. Therefore, additional samples of high-grade cancers, Gleason score 4+4 and higher were selected to enrich the number of training examples from high-grade prostate cancers.

It is a major undertaking in labor and cost to digitize pathology slides, and to digitize all benign or Gleason score 6 cases from the STHLM3 cohort will have extremely marginal (if any) effect on the performance of the training AI system. The bottleneck for training the system is the rarer, higher grade cases. This is the background to the selection procedure we used when digitizing the slides.

(2) The training data are from the same cohort. As a matter of fact, the corresponding author has published several papers using multi-center PCa cohorts. It will be more

convincing if this study can expand to other large cohorts (including different human races) for both training and testing.

We agree with the importance of external validation in large cohorts. This is the reason to why we performed external validation on the dataset from the Karolinska University Hospital (Test set 4). This analysis replicates the entire laboratory and digital pathology from an external data source with both a different laboratory and scanner compared to the training data. The advantage of this dataset is that it is assessed by the same pathologist (Professor Lars Egevad) as the training data, meaning that any differences between predictions on held-out data from the training dataset (Test set 1) and the external validation set (Test set 4) are likely to be due to differences in laboratory processing and digitalization (as opposed to differences in grading by different pathologists).

More external from additional sources are of course always valuable. We have therefore managed to access a dataset from Stavanger in Norway (n=1,220 slides), representing a large additional external test set. This test set has now been added to the manuscript.

(3) The n number for the atypical prostate tissues is small. The authors are suggested to include a larger number of atypical prostate tissue slides or to include PCa variants such as neuroendocrine prostate cancer or prostate cancer with neuroendocrine differentiation to test the application of conformal prediction.

We have extended the set of atypical prostate tissue with this revision. We have performed an additional pathology review and identified an additional set of n=152 cases of benign mimics and rare prostate cancer subtypes. We have digitized these new cases and combined them with the old Test set 5 that consisted of n=27 cases. Thus, the new updated Test set 5 includes n=179 cases in total. The results are similar after adding these additional 152 cases: Without conformal prediction, the AI system misclassifies 29% of these cases, while with conformal prediction it does not commit any errors but instead flags 80% of these unusual cases for human intervention.

Except from updating this test set, no modifications were done to the underlying AI-model providing the point predictions or the conformal predictor used to construct the prediction regions.

(4) Conformal prediction is not the only solution to assess the uncertainty of AI. The authors are suggested to compare or at least discuss different methods to demonstrate the advantage of their conformal prediction in AI-assisted histopathological diagnosis.

We of course agree with the reviewer that other methods for assessing reliability in AI predictions exist. The advantages with conformal prediction is that it is a mathematically very well developed theory with known and proved properties (for example, it is known from Vovk et al. [1] that conformal predictors are essentially the only way to achieve valid prediction regions). Conformal prediction thus works without having to rely on ad hoc or empirical evidence. In addition, conformal prediction is simple to implement and does not impose additional computational overhead. A new paragraph has been added to discussion

section of the manuscript, where we discuss other methods to assess the uncertainty of AI systems (page 11, line 260-267).

Vovk, V., Gammerman, A. & Shafer, G. Algorithmic learning in a random world. *Algorithmic Learn. a Random World* 1–324 (2005). doi:10.1007/b106715

(5) Please specify LE for the first time mentioned in the manuscript in line 93

This has been updated in the manuscript.

REVIEWERS' COMMENTS

Reviewer #1 (Remarks to the Author):

The authors have addressed my concerns. However, I do not agree with their statement that "The code used for training the underlying deep learning models has a large number of dependencies on internal tooling, infrastructure and hardware, and its release is therefore not feasible.". Other groups have solved this problem, e.g., please check the Warwick Digital Pathology Lab's workflows, which are fully publicly released as the "tiatoolbox", or Mahmood Labs workflows, which are fully publicly released as the CLAM pipeline. Although the current manuscript can be accepted despite this shortcoming, I urge the authors to adopt such a transparent approach for their full analysis pipeline in the future.

Reviewer #3 (Remarks to the Author):

The authors have addressed my concerns. I have no further questions.